# Integron activity accelerates the evolution of antibiotic resistance

**Célia Souque[1]\*, José Antonio Escudero[1,2]†, R Craig MacLean[1]†**

[1]University of Oxford, Department of Zoology, Oxford, United Kingdom; [2]Universidad Complutense de Madrid, Departamento de Sanidad Animal and VISAVET, Madrid, Spain

**Abstract** Mobile integrons are widespread genetic platforms that allow bacteria to modulate the expression of antibiotic resistance cassettes by shuffling their position from a common promoter. Antibiotic stress induces the expression of an integrase that excises and integrates cassettes, and this unique recombination and expression system is thought to allow bacteria to 'evolve on demand' in response to antibiotic pressure. To test this hypothesis, we inserted a custom three-cassette integron into *Pseudomonas aeruginosa* and used experimental evolution to measure the impact of integrase activity on adaptation to gentamicin. Crucially, integrase activity accelerated evolution by increasing the expression of a gentamicin resistance cassette through duplications and by eliminating redundant cassettes. Importantly, we found no evidence of deleterious off-target effects of integrase activity. In summary, integrons accelerate resistance evolution by rapidly generating combinatorial variation in cassette composition while maintaining genomic integrity.

**\*For correspondence:**
celia.souque@gmail.com

†These authors contributed equally to this work

**Competing interests:** The authors declare that no competing interests exist.

## Introduction

Given the mounting threat posed by antibiotic resistance, we need a better understanding of the mechanisms used by bacteria to evolve resistance to antibiotics. Mobile integrons (MIs) are widespread elements providing a platform for the acquisition, shuffling, and expression of gene cassettes, many of which are antibiotic resistance genes (*Recchia and Hall, 1995*; *Escudero et al., 2015*). These elements are typically associated with transposons and conjugative plasmids and have played an important role in the evolution of resistance in pathogenic bacteria (*Partridge et al., 2018*). Five classes of MIs, based on the sequence of their integrase, have arisen independently through association with diverse mobile elements, but the class 1 integron is, by far, the most prevalent and clinically relevant. The first multidrug resistance plasmids that were isolated in the 1950s carried class 1 MIs (*Liebert et al., 1999*; *Mitsuhashi et al., 1961*; *Rownd et al., 1966*; *Stokes and Hall, 1989*), and recent surveys have shown that class 1 integrons are found in a substantial fraction of isolates of *Escherichia coli* (*Halaji et al., 2020*; *Rao et al., 2006*; *Yu et al., 2003*), *Klebsiella pneumoniae* (*Firoozeh et al., 2019*; *Li et al., 2013*), *Pseudomonas aeruginosa* (*Oliver et al., 2015*; *Ruiz-Martínez et al., 2011*), and *Acinetobacter baumanii* (*Chen et al., 2015*; *Turton et al., 2005*).

Mobile integrons consist of an integrase encoding gene named *intI* followed by a recombination site, *attI* (*Hall et al., 1991*; *Partridge et al., 2000*) and a variable array of mobile gene cassettes (typically 2–5 in mobile integrons) ending each in a characteristic hairpin recombination site called the *attC* site (*Hall et al., 1991*). Integron cassettes usually lack a promoter, and their expression is driven by the Pc promoter located upstream of the array (*Collis and Hall, 1995*), such that expression levels are highest for cassettes closest to the promoter (*Collis and Hall, 1995*). The SOS response (named after the Morse code ...—...) induces the expression of integrases (*Guerin et al., 2009*), which then allows for the efficient integration and excision of cassettes in the array through *attC* × *attI* and *attC* × *attC* reactions, respectively (*Collis and Hall, 1992*). A

**eLife digest** From urinary tract infections to bacterial pneumonia, many diseases can now be treated through a course of antibiotics. Yet bacteria have evolved to respond to this threat, gaining new antibiotic resistance genes that allow them to evade the drugs. Addressing this growing issue requires to either discover new antibiotics, or to stop resistance before it emerges – a strategy that can only work if scientists know exactly how this mechanism takes place.

For bacteria, it is a waste of resources to produce the proteins that confer resistance if antibiotics are absent. In fact, doing so can decrease their chance to survive and reproduce. A genetic element known as an integron can help to manage that burden. This piece of genetic information is formed of a succession of 'cassettes' containing antibiotic resistance genes. More proteins are made from the genes present at the start of the integron, compared to the ones towards the end. When bacteria encounter antibiotics, an enzyme called integrase is activated, allowing the organisms to shuffle the order of their cassettes in the integron. It is thought – but not yet proven – that this mechanism helps bacteria to activate their resistance 'on demand'.

To find out, Souque et al. engineered the bacteria *Pseudomonas aeruginosa* to carry a custom integron with three cassettes, each helping the organism to resist to a different antibiotic. In addition, only half of the bacteria had a working integrase and could therefore shuffle their gene cassettes. The organisms were then exposed to an increasing amount of the antibiotics for which the cassette in the last position provided resistance. The bacteria with a working integrase survived longer than those without, as they were able to shuffle their cassettes and move the useful antibiotic resistance gene into top position. In addition, the cassettes carrying the genes to resist to other types of antibiotics were excised from the genetic information and lost.

Understanding integrons could guide future antibiotic treatment strategies, for instance by combining antibiotics with chemicals that block integrase activity. It might also be possible to force bacteria to delete resistance cassettes by cycling through different antibiotics.

peculiarity of this system is that integron recombination is semi-conservative, as only the bottom strand of the cassette is excised from the array following a recombination pathway that includes a replication step (*Loot et al., 2012*). The implication of this is that cassette excision and re-integration can lead to two different results: either a 'cut and paste' outcome, resulting in the movement of a cassette within an array, or a 'copy and paste' outcome, leading to the insertion of a duplicate copy of a cassette in the conserved array (*Escudero et al., 2015*). An overview of the mechanisms of integron activity is presented in *Figure 1a*.

Due to the stress-inducible regulation of integrase activity, integrons have been proposed to accelerate bacterial evolution by providing 'adaptation on demand' (*Escudero et al., 2015*). According to this hypothesis, integrase-mediated cassette re-shuffling in stressful environments allows bacteria to optimize cassette expression and maximize fitness: useful cassettes can be brought forward to ensure maximal expression, while unnecessary cassettes can be kept at the end of the array as a low cost memory of once-adaptive functions, ready to be brought forward when needed (*Escudero et al., 2015*). Stress-inducible regulation also helps to minimize the costs associated with integrase expression (*Lacotte et al., 2017*; *Starikova et al., 2012*), which are thought to come from increased genomic instability created by off-target integrase activity (*Harms et al., 2013*). Although antibiotics have diverse modes of action, many of the most common classes of antibiotic cause DNA damage that induces the SOS response (*Kohanski et al., 2010*). This link between antibiotic exposure and integrase activity suggests that cassette re-shuffling may allow pathogenic bacteria to rapidly adapt to novel antibiotic challenges.

While the molecular mechanisms of integron shuffling are known in detail, our ability to understand the evolutionary benefits provided by this fascinating genetic platform is limited by our understanding of the population biology of integron-mediated antibiotic resistance. For example, constitutive over-expression of the integrase enzyme has been shown to accelerate the evolution of chloramphenicol resistance through the loss of cassettes between Pc and the resistance cassette as well as the formation of co-integrates between integron copies (*Barraud and Ploy, 2015*). However, to the best of our knowledge, the benefits of cassette shuffling under the integrase natural promoter

and its associated LexA regulation have never been investigated. This is an important limitation, as parameters such as the re-insertion rate of excised cassettes and fitness costs of integrase expression are predicted to have a strong impact on the evolutionary benefits of the integrase (*Engelstädter et al., 2016*). Moreover, integron cassette shuffling has rarely been studied in the large, natural plasmids where class 1 integrons often occur.

Here we directly test the 'adaptation on demand' hypothesis using experimental evolution in populations of *P. aeruginosa* carrying a variant of the broad host range plasmid R388. We replaced the naturally occurring class 1 mobile integron in R388 with a customized integron containing three antibiotic resistance cassettes: *dfrA5* (a trimethoprim resistant dihydrofolate reductase [*Sundström et al., 1988*]), *bla*$_{VEB-1}$ (an extended-spectrum-β-lactamase [*Poirel et al., 1999*]), and *aadB* (an aminoglycoside-2′-adenylyltransferase [*Cameron et al., 1986*]). To directly investigate the role of integrase activity in this system, we also generated a truncated integrase mutant that allows normal levels of cassette expression, but not recombination. Using this system, we found that integrase activity leads to rapid and extensive cassette re-shuffling in response to strong selection for

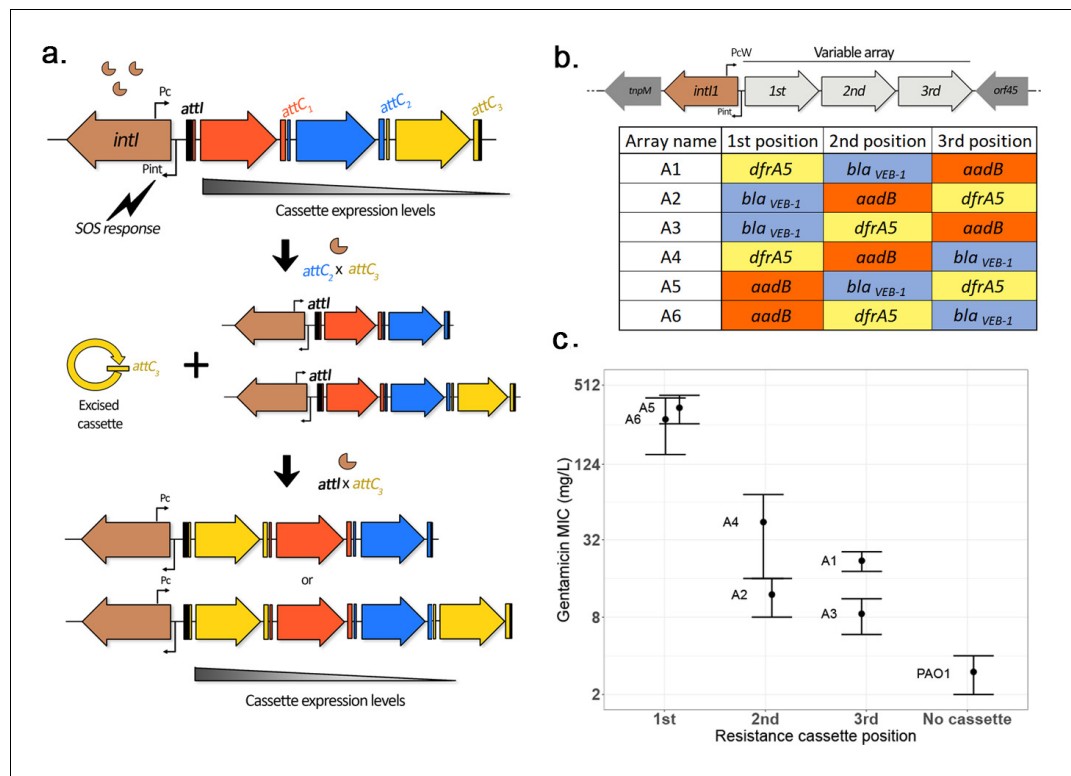

**Figure 1.** Overview of integron system. (a) Diagram of the integron mechanism: the integron consists of an integrase gene, *intI*, followed by an array of promoterless gene cassettes (represented here by arrows). Cassettes are expressed from the Pc promoter within the integrase gene, with decreasing cassette expression along the array. Following the induction of the SOS response, the integrase enzyme promotes cassette excision (recombination between a cassette *attC* site and the *attC* of the preceding cassette, causing excision of the cassette into its circular form). Due to the presence of a replication step in the excision process, a copy of the original array is conserved. Re-integration of the circular cassette can then occur through recombination between the cassette *attC* site and the *attI* site located at the start of either array, leading to an apparent 'cut-and-paste' recombination if the cassette integrates in the excised array, or can be assimilated to a 'copy-and-paste' outcome if it integrates in a conserved copy of the array. By shuffling and duplicating cassettes, the integron has the potential to quickly modulate cassette expression levels. (b) Custom integron arrays: the native integron array of the R388 plasmid was replaced by the custom integron arrays WTA1 to WTA6 containing three integron cassettes in every possible order. (c) Effect of position of the *aadB* cassette on gentamicin resistance levels. Error bars represent standard error (n = 2–4).

The online version of this article includes the following figure supplement(s) for figure 1:

**Figure supplement 1.** Transcriptional and translational origin of the *aadB* expression gradient.

increased gentamicin resistance. Specifically, integrase activity caused the insertion of duplicate copies of *aadB* cassettes in the first position of the integron, followed by the loss of redundant cassettes. Crucially, this accelerated the ability of populations to adapt to antibiotic stress, providing good support for the 'adaptation on demand' hypothesis. Finally, we show that rapid duplications can also occur with *bla*$_{VIM-1}$ cassettes, which confer resistance to 'last line of defense' carbapenem antibiotics, in a recently isolated clinical plasmid under meropenem selection.

## Results

### A combinatorial, three-cassette integron system to investigate the impact of cassette position

We replaced the naturally occurring class 1 integron of plasmid R388 with all six possible configurations of a class 1 integron containing three antibiotic resistance cassettes, including *dfrA5*, *aadB*, and *bla*$_{VEB-1}$ and transformed our integron variants into *P. aeruginosa* PA01 (*Figure 1b*). Integrons have played an important role in the evolution of antibiotic resistance in the opportunistic pathogen *P. aeruginosa* and are highly prevalent in *P. aeruginosa* high-risk clones (*Oliver et al., 2015*).

As expected, resistance levels to gentamicin declined as the *aadB* cassette moved further away from the integrase (*Figure 1c*). Interestingly, the relationship between *aadB* position and resistance was not linear: we observed a 6- to 24-fold difference in minimum inhibitory concentration (MIC) between arrays containing *aadB* in first and second positions, but a less than twofold difference between arrays with *aadB* in second and third places. In order to investigate the mechanisms behind this trend, we measured the *aadB* cassette transcription levels of the different arrays. Instead of a sharp drop, we observed a steady decrease depending on the cassette position (*Figure 1—figure supplement 1a*). Previous work has shown that two short open-reading frames contained within the *attI* site can substantially enhance the translation of a cassette lacking a Shine–Dalgarno (SD) sequence when the cassette is located in first position (*Hanau-Berçot et al., 2002*; *Papagiannitsis et al., 2017*), showing that cassette position can also modulate translation levels (*Hanau-Berçot et al., 2002*; *Jacquier et al., 2009*). Interestingly, the *aadB* cassette contains a reduced SD box (*Figure 1—figure supplement 1b*), suggesting that the steep gradient in gentamicin resistance between first and second positions was mostly due to decreased translation.

### Integrase activity accelerates the evolution of antibiotic resistance

Given the strong effect of *aadB* cassette position on gentamicin resistance, we decided to use this combination of cassette and antibiotic to experimentally test the hypothesis that integrase activity accelerates resistance evolution. To properly measure the effect of integrase activity on evolvability, we constructed a Δ*intI1* mutant of the A3 array lacking 818 bp of *intI1* (total length is 1014 bp) but conserving the Pc and Pint promoters. We challenged independent populations of WTA3 and Δ*intI1*A3 with increasing doses of gentamicin using an 'evolutionary ramp' design (*Gifford et al., 2018*; *San Millan et al., 2016*). Importantly, we did not detect any difference in initial gentamicin resistance (MICexp = 24 mg/L) between strains with array A3 or the Δ*intI1*A3 mutant in the conditions of the evolution experiment (see Materials and Methods). We passaged 65 populations of each strain, starting at 1/8 MIC (i.e. 3 mg/L) and doubling the concentration of gentamicin each day until reaching 1024 times (24.5 g/L) the initial MIC (*Figure 2a*). As controls, 15 populations of each strain were passaged without antibiotic (no selection for gentamicin resistance), while 15 populations were passaged at a constant dose of 1/8 MIC (3 mg/L) to generate weak selection for gentamicin resistance and plasmid maintenance.

The rapid increase in antibiotic concentration in the 'evolutionary ramp' treatment ensures that populations must either evolve increased resistance or face extinction (once concentrations exceed the MIC of the parental strains). Given this, measuring the rate at which populations go extinct provides a way to measure the evolvability of a strain. Crucially, populations of WTA3 populations with a functional integrase had a higher survival rate than those of the Δ*intI1*A3 mutant, showing that the integrase can increase evolvability for antibiotic resistance (*Figure 2b*; log-rank test: Chisq = 17.7, df = 1, p=3e-05). We did not detect any extinctions in either of the controls, showing that the higher extinction rate of Δ*intI1*A3 populations was driven by exposure to high doses of gentamicin. We confirmed the evolution of high level of gentamicin resistance by measuring the MICs of a subset of

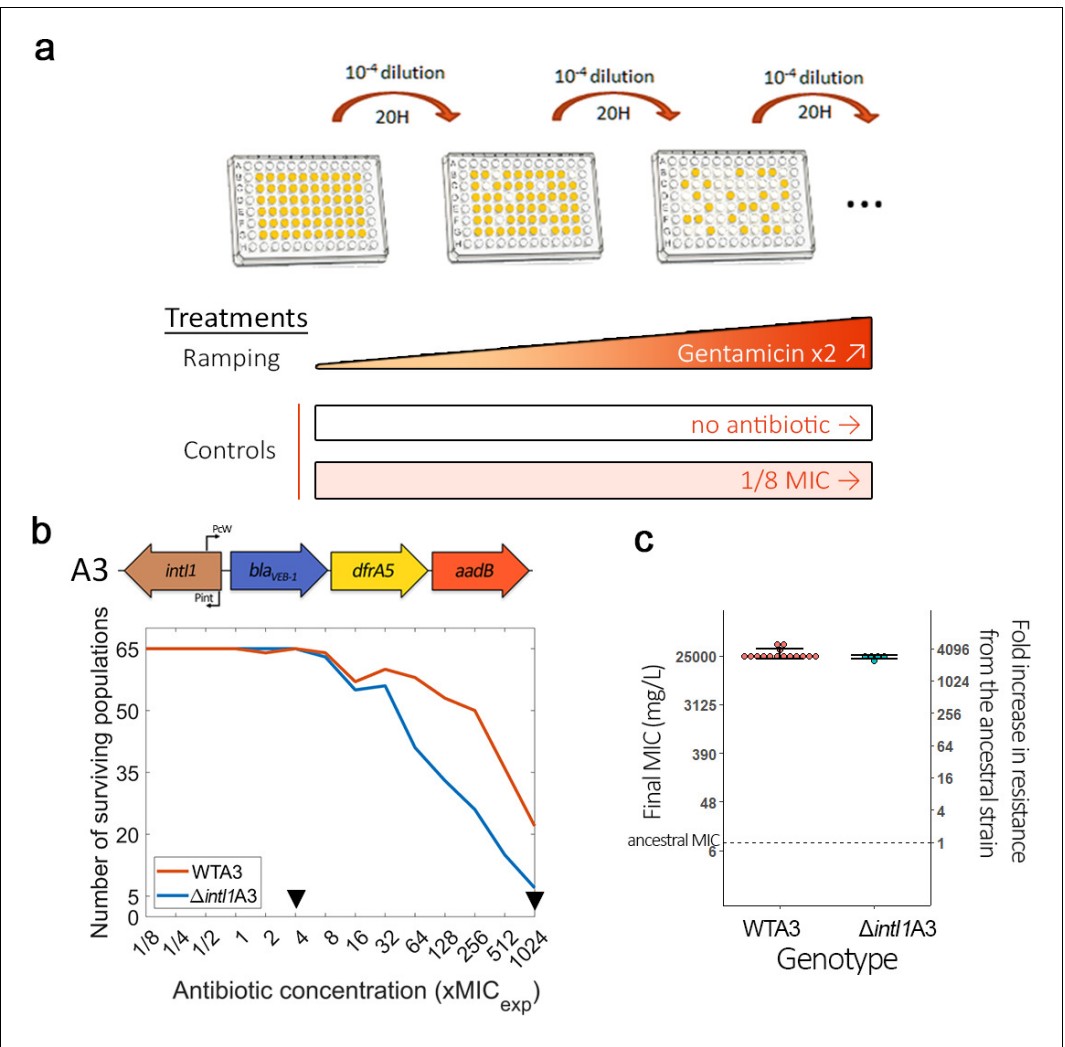

**Figure 2.** Integrase activity can increase bacteria evolvability against antibiotics. (a) Schematic representation of the experimental evolution protocol. (b) Top: Representation of the WTA3 integron. Bottom: Survival curves of the PA01:WTA3 and PA01:ΔintI1A3 populations during ramping treatment, monitored using OD595. The black triangles represent time points where populations were sequenced using whole genome sequencing. (c) Average minimum inhibitory concentration (MIC) of a random subset of the PA01:WTA3 (15 populations) and PA01:ΔintI1A3 populations (five populations) from the final ×1024 MIC time point. The MIC of the ancestral populations is represented by a dashed line. Error bars represent standard deviation. The MIC of each population was averaged from three biological replicates.

populations from the final time point (*Figure 2c*). We observed a similar level of resistance between ΔintI1A3 (mean MIC = 24,200 mg/L; s.d. = 1864; n = 5) and WTA3 (mean MIC = 27,800 mg/L; s. d. = 6031; n = 15) populations (t = 2.044, p=0.056), showing that integrase activity increased the likelihood of resistance evolution, but did not impact the final resistance levels of surviving populations. To understand the mechanisms by which integrase activity accelerates evolution, we sequenced DNA extracted from randomly chosen populations at a mid-point of the experiment (×4 MIC; n = 24 WTA3 and 22 ΔintI1A3) and all populations that survived until the end of the experiment (×1024 MIC; n = 21 WTA3 and 6 ΔintI1A3 populations).

## Integron evolution under antibiotic treatment

We found evidence for widespread cassette re-arrangement in WTA3 populations and identified four novel integron structures that were formed by insertion of the *aadB* cassette and/or deletions of *bla*$_{VEB-1}$-*dfrA5* (*Figure 3a,b* and *Supplementary files 2* and *3*). The junction sites for cassette

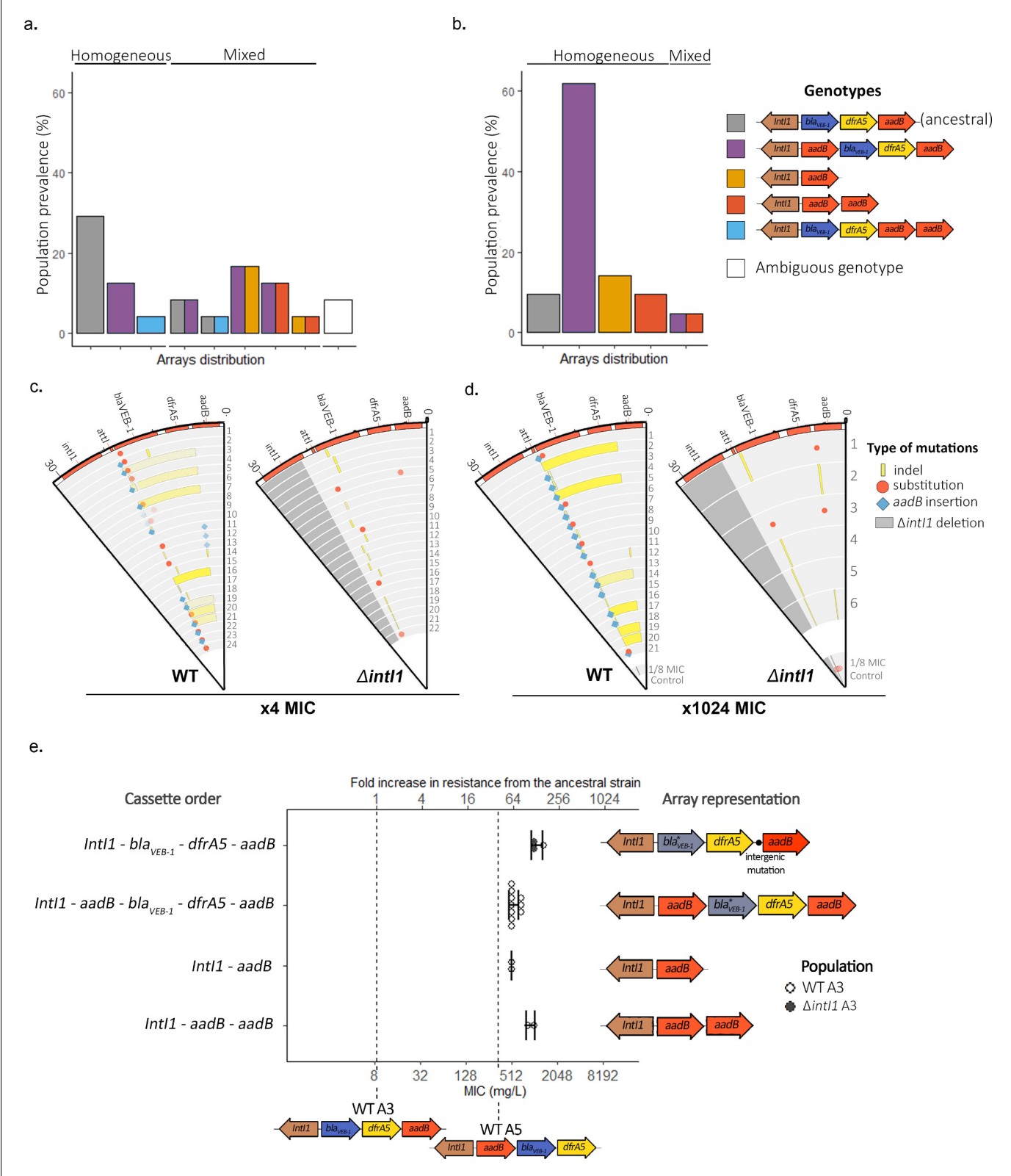

**Figure 3.** Extensive cassette re-arrangements are linked with integrase activity. (**a,b**) Distribution of cassette re-arrangements at ×4 MIC (**a**) and ×1024 MIC (**b**) time points in the WTA3 populations. Homogeneous populations represent populations where only one type of array could be identified while mixed populations contain different arrays as indicated by the corresponding colors. Ambiguous populations correspond to re-arrangements that could not be identified with confidence from short-read data. No re-arrangement was found in the Δintl1A3 populations. (**c,d**)

*Figure 3 continued on next page*

*Figure 3 continued*

Representation of the plasmid mutations and re-arrangements in the surviving PA01:WTA3 and PA01: Δ*intI1*A3 populations at ×4 MIC (**c**) and ×1024 MIC (**d**), mapped to the integron reference sequence. Each circle represents a separate population, with the inner circle representing the variants present in an equimolar pool of six 1/8 MIC control populations. Indels are represented in yellow and single-nucleotide substitutions in red. *aadB* insertions are represented by blue lozenges. The color intensity represents the frequency of the corresponding mutation/recombination. The dark gray area in the PA01:Δ*intI1*A3 populations represents the location of the *intI1* deletion. (**e**) Resistance levels provided by evolved plasmids in the ancestral chromosomal background. The plasmids of 15 PA01:WTA3 and 2 PA01:Δ*intI1*A3 populations were extracted and transformed back into the ancestral PA01 strain. The populations are grouped by array cassette order. Other mutations in the arrays (*bla*$_{VEB-1}$ mutations or intergenic mutations upstream of *aadB*) are indicated in the arrays representations on the right. Resistance levels of the less (WTA3) and most (WTA5) resistant unevolved custom arrays are represented by the dashed lines. Error bars represent standard deviation and the MIC of each plasmid was averaged from three biological replicates.

The online version of this article includes the following figure supplement(s) for figure 3:

**Figure supplement 1.** Plasmid mutations and re-arrangements at (**a**) ×4 MIC and (**b**) ×1024 MIC.
**Figure supplement 2.** Re-arrangements detection by PCR in ×1024 MIC WTA3 populations.
**Figure supplement 3.** Re-arrangements in the plasmid backbone of the WT populations at ×1024 MIC.

insertions and deletions were consistent with integrase activity: recombination happened at the 5′-GTT-3′ triplet of the *attI1*, *attC*$_{aadB}$, and *attC*$_{dfrA5}$ sites. We did not find any evidence for cassette re-arrangements in Δ*intI1*A3 populations or in control WTA3 populations that we selected at a low dose of gentamicin (×1/8 MIC), while the entire R388 plasmid was lost in all Δ*intI1*A3 and WTA3 populations passaged without antibiotic. Cassette re-arrangements were found in most populations at the ×4 MIC time point, and approximately 90% of populations (19/21) contained cassette re-arrangements by the end of the experiment, highlighting the importance of integrase activity in resistance evolution. Integron structural polymorphisms were found in 50% of populations (12/24) at ×4 MIC, but this within-population diversity was transient and almost all populations contained a single dominant integron structure by the final time point.

The most common novel integron structure contained a 'copy and paste' insertion of *aadB* in first position via *attI* × *attC*$_{aadB}$ recombination (ie *aadB-bla*$_{VEB-1}$*dfrA5aadB*). We measured the impact of this novel array on gentamicin resistance level by transferring the evolved plasmids into the ancestral chromosomal background (**Figure 3e**). This novel integron is associated with a large increase (64-fold) in gentamicin resistance due to the dominant effect of first position on *aadB*, with similar levels to our constructed arrays with *aadB* in first position (**Figure 3e**). Interestingly, we did not identify any *aadBbla*$_{VEB-1}$*dfrA5* arrays, which would be the result of an *aadB* excision followed by reintegration of *aadB* in first position within the same array, highlighting the prevalence of 'copy and paste' cassette insertions. Degenerate integrons that lack the *bla*$_{VEB-1}$ and *dfrA5* cassettes (i.e. either *aadB* or *aadBaadB* arrays) were also present at relatively high frequency at both the ×4 and ×1024 MIC time points. Interestingly, in mixed arrays populations, these reduced arrays were always observed in conjunction with the *aadB-bla*$_{VEB-1}$*dfrA5aadB* array and never with the ancestral array. This repeated association provides good evidence that degenerate arrays evolved via *aadB* insertion in first position, to form the common *aadB-bla*$_{VEB-1}$*dfrA5aadB* array, followed by the en bloc deletion of the other cassettes (*bla*$_{VEB-1}$*dfrA5aadB* or *bla*$_{VEB-1}$*dfrA5*) to form *aadB* and *aadBaadB* arrays. Recombination happened at the 5′-GTT-3′ triplet of the R box of *attC*$_{aadB}$ and *attC*$_{dfrA5}$ sites, suggesting that these deletions were driven by integrase activity, although we cannot rule out the possibility that the *bla*$_{VEB-1}$*dfrA5aadB* cassette deletion was driven by homologous recombination between *aadB* cassettes. The relative prevalence of these two degenerate arrays did not change between the ×4 and ×1024 MIC time points (four *aadB* against three *aadBaadB* arrays at ×4 MIC and three *aadB* against three *aadBaadB* arrays at ×1024 MIC), which suggests that the second *aadB* cassette in the *aadBaadB* array is redundant. In line with this argument, we found a marginal difference (×2) in MIC between evolved plasmids carrying *aadB* (mean = 512, s.d. = 0) and *aadBaadB* (mean = 939, s.d. = 121). Finally, we found arrays containing a duplicate copy of *aadB* at the end of the array (*bla*$_{VEB-1}$*dfrA5aadBaadB*), which are likely to have been formed by the insertion of an *aadB* cassette in the middle or at the end of the array through the less frequent *attC* × *attC* integration (intermolecular) reaction. These arrays were only found at the ×4 MIC time point, strongly suggesting that they conferred a small increase in gentamicin resistance that was ultimately an evolutionary dead end under strong selection for elevated resistance.

In addition to changes in integron structure, we found widespread integron evolution by mutations in both the WTA3 and $\Delta intI1$A3 populations. Mutations in $bla_{VEB-1}$ were found in more than 80% of WTA3 and $\Delta intI1$A3 populations from the $\times 4$ MIC time point, and in almost all populations where the $bla_{VEB-1}$ cassette was maintained at the $\times 1024$ MIC time point, including 5/6 $\Delta intI1$A3 and 16/16 WTA3 populations. All mutations in $bla_{VEB-1}$ were amino acid substitutions (n = 9) or indels (n = 16) and the 23 amino acid signal peptide was a hotspot for mutations (12 of 25 $bla_{VEB-1}$ mutations), suggesting strong selection to eliminate the secretion of this redundant resistance protein (*Supplementary files 2* and *3*). Furthermore, similar $bla_{VEB-1}$ mutations were also found in the ⅛ MIC controls, demonstrating that these mutations were beneficial under low doses of gentamicin, as we would expect if this cassette imposed an important fitness cost. It is unclear if this cost of $bla_{VEB-1}$ was driven by the presence of gentamicin (i.e. collateral sensitivity) because the entire R388 plasmid was lost in every control population that was passaged in antibiotic-free medium. Parallel evolution also occurred close to the putative translation initiation site of the *aadB* cassette. These mutations were very rare at the $\times 4$ MIC time points (2/46 populations), but were present at a high frequency in $\Delta intI1$A3 populations from the final time point (4/6 populations) and are linked with high level of gentamicin resistance (*Figure 3e*). We speculate that these mutations were favored in $\Delta intI1$A3 populations as they increased the translation rate of the weakly expressed third position *aadB* cassette and offer an alternative mechanism to increase *aadB* expression in the absence of re-arrangements. Similarly, we identified one 41 bp deletion within the *dfrA5 attC* site of a WTA3 population, which may increase translational coupling with the previous *dfrA5* cassette (as in *Jacquier et al., 2009*) or lead to the creation of a fused protein with part of the previous cassette. Finally, we observed similar extended deletions in one WTA3 population and in the 1/8 MIC WTA3 pooled control. These deletions occur between $attC_{aadB}$ and different positions within the plasmid *trwF* gene, effectively deleting most of the genes involved in mating pore formation (*Figure 3—figure supplement 1*), with the sequence of the junction sites pointing toward potential off-target activity of the integrase (abundance of 5'-GNT-3' secondary sites) (*Figure 3—figure supplement 3*).

## Chromosomal evolution

The integron integrase is known to have off-target effects, suggesting that integrase activity may also have an important effect on chromosomal evolution by reducing genomic stability through recombination between chromosomal pseudo-*attC* or pseudo-*attI* sites, leading to an increase in deletions and re-arrangements (*Harms et al., 2013*).

Chromosomal evolution occurred more rapidly in the $\Delta intI1$A3 populations than in the WTA3 populations, as demonstrated by the high cumulative frequency of mutations in $\Delta intI1$A3 populations (mean = 1.01; s.e. = 0.17, see *Figure 4—figure supplement 2*) at $\times 4$ MIC compared to WTA3 populations (mean = 0.48; s.e. = 0.11; t = −2.69, df = 35.83, p=0.01, Welch two-sample t-test). However, accelerated chromosomal evolution in the absence of integrase activity was short-lived, and cumulative frequency of mutations in the WTA3 and $\Delta intI1$A3 populations was almost identical at the end of the experiment (mean WTA3 = 2.45 SNPs; mean $\Delta intI1$A3 = 2.65 SNPs, see *Figure 4—figure supplement 1*). Crucially, we found only one case of chromosomal recombination, with a 3.5 kb deletion between the two highly homologous ccoN1 and ccoN2 cytochrome C subunits in one WTA3 population (*Supplementary file 2*), showing that off-target effects of the integrase were undetectable in our experiment, in spite of our extensive genomic sequencing.

In total, we identified 41 different SNPs and 58 short indels in eight intergenic regions and 41 genes, with a similar spectrum of mutations in the ramping WTA3 and $\Delta intI1$A3 populations (*Figure 4*; *Supplementary files 2* and *3*). Several lines of evidence indicate that the overwhelming majority of mutations were beneficial mutations that reached high frequency as a result of selection. First, many of the mutated genes have known roles in antibiotic resistance; for example, 11/41 mutated genes have also been identified in an aminoglycoside resistance screen in *P. aeruginosa* (*Schurek et al., 2008*). Second, parallel evolution was very common. Repeated evolution occurred in 11 of 42 (26%) genes and 3 of 8 (38%) intergenic regions with 68% of mutations occurring in these genes. Only 1 of 41 mutations in coding regions was synonymous, providing evidence that the rapid evolution of proteins was driven by positive selection, and not simply by an elevated mutation rate. Finally, we found almost no overlap between the genes that were mutated in the ramping populations and the controls, implying that the evolutionary response of the ramping populations was dominated by selection for high levels of gentamicin resistance. For both control regimens, the

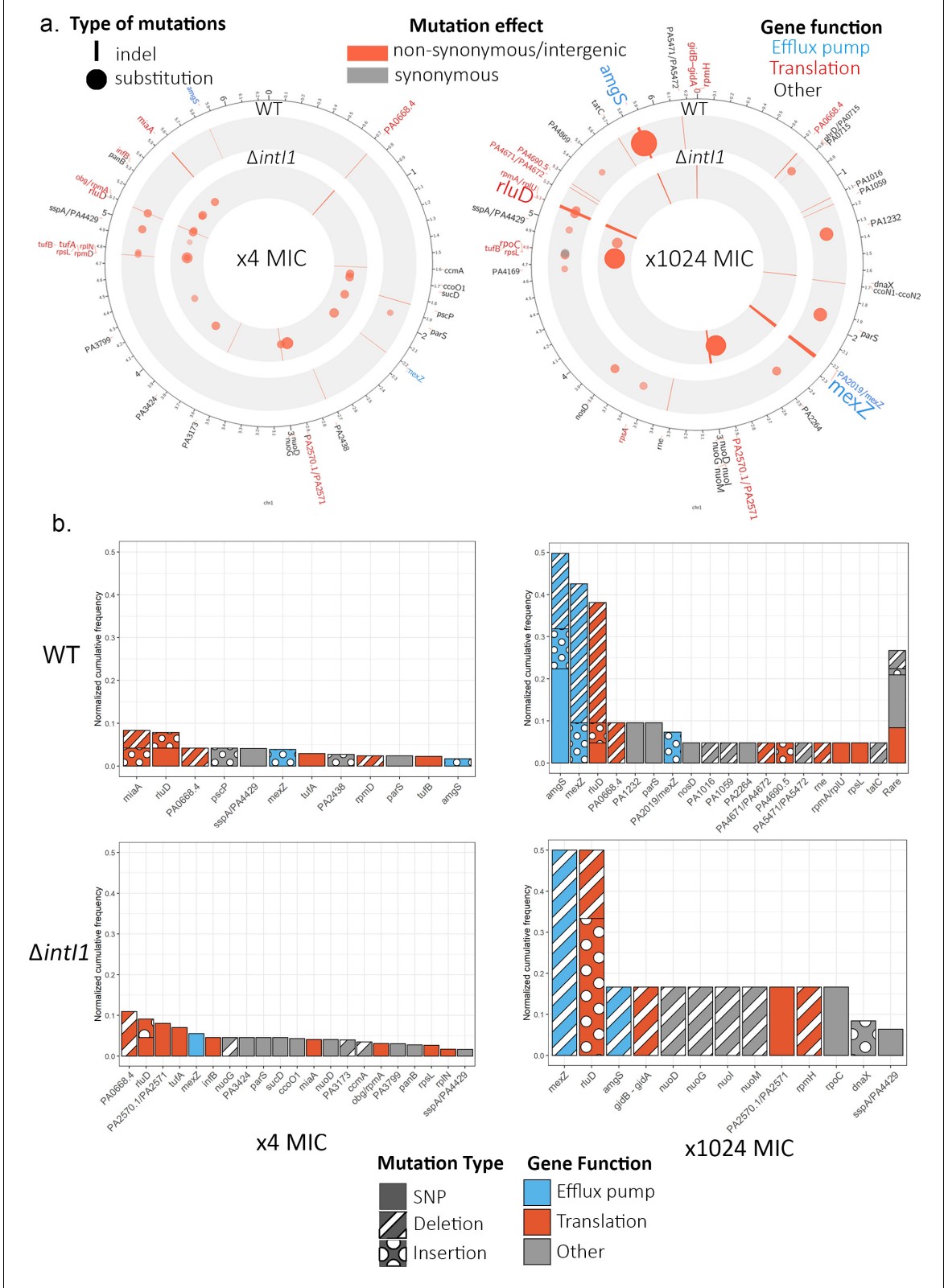

**Figure 4.** Chromosome evolution. (**a**) Summary of the chromosomal mutations at ×4 MIC (left) and at ×1024 MIC (right) mapped to the PAO1 reference sequence. Each circle represents a summary of each genotype. The type (indel, substitution) of mutation for each gene is represented by the shape of the marker (line, circle), while the marker color represents the effect of the mutations (nonsynonymous/intergenic vs. synonymous), and its color intensity and size represent its normalized cumulative frequency per gene. The size of the gene labels on the outer ring represents the overall

*Figure 4 continued on next page*

*Figure 4 continued*

cumulative frequency of mutations present in this gene across all populations from this time point. (b) Cumulative frequency of mutations for each gene normalized by the number of populations within each genotype and time point. Genes are colored by resistance mechanism, and the type of the mutations (single-nucleotide substitution, insertion, deletion) is indicated by the patterning.

The online version of this article includes the following figure supplement(s) for figure 4:

**Figure supplement 1.** Summary statistics of mutations in the ×1024 MIC populations.
**Figure supplement 2.** Summary statistics of mutations in the ×4 MIC populations.

chromosomal genes with the most mutations were *cdrA* (PA4625), involved in biofilm formation (*Reichhardt et al., 2018*), and PA1874, a hypothetical protein. Similarly to the ramping populations, no wide-scale chromosomal re-arrangements were observed, and no major differences could be found between the distribution of mutations between WTA3 and Δ*intl1*A3 populations in both control regimens at x1024 MIC (*Figure 4—figure supplement 1*).

In the ramping populations, the initial stages of adaptation to gentamicin were driven by mutations in a very diverse set of genes, with a strong bias toward genes that are involved in translation, such as *rluD* and PA0668.4, which encodes for the 23S ribosomal RNA (*Figure 4*). Interestingly, we observed divergent mutational trajectories of evolution in the WTA3 and Δ*intl1*A3 backgrounds: the number of genes that were mutated in both backgrounds was small (n = 7) relative to the total number of mutated genes in either background (n = 26) and the correlation in mutation frequencies across backgrounds was weak (rho = 0.029; p=0.89 Spearman test).

Continued selection for elevated gentamicin resistance resulted in two changes in chromosomal evolution (*Figure 4*). First, chromosomal evolution became dominated by mutations in a few key target genes, implying that many of the trajectories of chromosomal evolution followed during early adaptation ultimately led to evolutionary dead ends. For example, the correlation in mutation frequencies between early and late time points was very weak, in both WTA3 (rho = −0.10, p=0.58) and Δ*intl1*A3 (rho = 0.20, p=0.307). In particular, we found evidence of extensive parallel evolution in *mexZ*, *amgS*, and *rluD* in both the WTA3 and Δ*intl1*A3 populations. At a functional level, the mutations found at ×1024 MIC are predominantly involved in antibiotic efflux, as opposed to translation. *mexZ* is a transcription factor that represses the expression of the *mexXY* multidrug efflux pump operon. Mutations inactivating *mexZ* cause a 2- to 16-fold increase in aminoglycoside resistance and have been widely identified in aminoglycoside-resistant *P. aeruginosa* isolates found in cystic fibrosis patients (*Vogne et al., 2004*). AmgS is part of an envelope stress-responsive two-component system AmgRS (*Lau et al., 2013*), and *amgS* mutations upregulate the *mexXY* multidrug efflux system in the presence of aminoglycosides (*Lau et al., 2015*).

## Cassette duplication in a clinically relevant plasmid

Resistance to carbapenem antibiotics in *P. aeruginosa* has emerged as an important clinical threat; for example, the WHO has designated carbapenem-resistant *P. aeruginosa* as a 'critical priority' for the development of new antibiotics. Interestingly, mobile integrons carrying multiple *bla*<sub>VIM-1</sub> carbapenemase cassettes have been found in clinical isolates of *P. aeruginosa* (*San Millan et al., 2015b*), suggesting that cassette duplications may play an important role in clinical settings. To test this idea, we challenged 30 populations of *P. aeruginosa* carrying a plasmid (pAMBL-1), which contains an integron carrying a single copy of *bla*<sub>VIM-1</sub> followed by *aadB*, with increasing doses of meropenem using a similar evolutionary ramp experiment (*Figure 5*). PCR screening of populations that survived at ×2 MIC identified numerous cassette re-arrangements of both the *bla*<sub>VIM-1</sub> and *aadB* cassettes, with potential *bla*<sub>VIM-1</sub> duplications occurring in all 14 surviving populations (*Figure 5*). Short-read sequencing of clones isolated from two of these populations confirmed the presence of duplications, as demonstrated by increased copy number of *bla*<sub>VIM-1</sub> per plasmid (2.0 copies/plasmid [95% CI: 1.90–2.10] and 2.67 copies/plasmid [95% CI: 2.53–2.78]). Although it is not possible to definitely prove the role of the integrase without control populations lacking a functional integrase, these results strongly support the idea that 'copy and paste' outcomes of cassette re-arrangements can drive the rapid evolution of elevated carbapenem resistance and was the mechanism behind the *bla*<sub>VIM-1</sub> amplification observed in the plasmid pAMBL2 isolated in the same hospital (*San Millan et al., 2015b*).

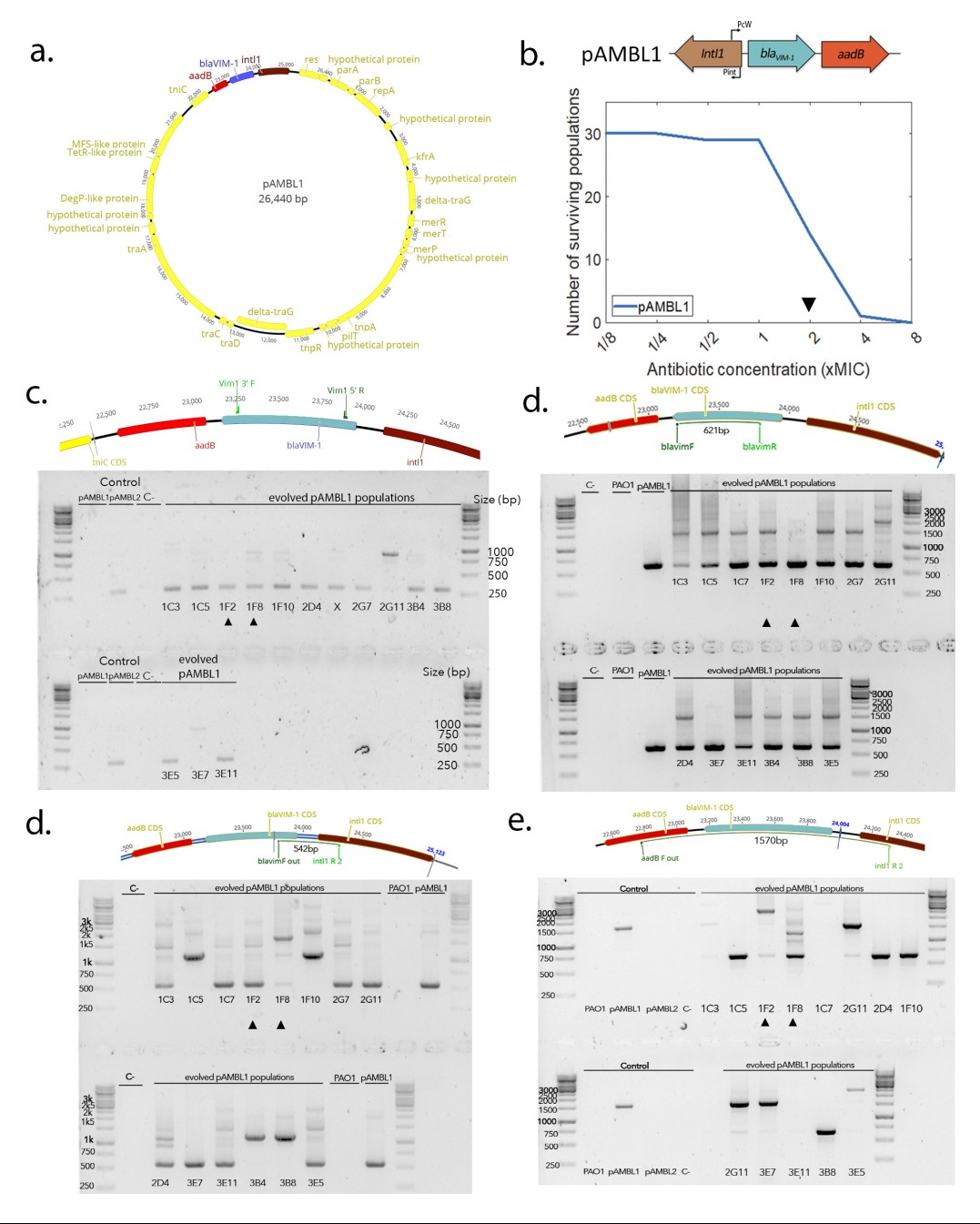

**Figure 5.** pAMBL1 re-arrangements. (a) Representation of the pAMBL1 plasmid. The integron is highlighted in color. (b) Survival curve of the 30 PA01: pAMBL1 populations under ramping treatment. The black arrow indicates the time point at which populations were plated and duplications amplified by PCR/NGS. (c–e) Detection of cassette re-arrangements by PCR. The expected positions of the primers on the ancestral pAMBL1 and the size of the corresponding amplicon are indicated on top.

## Discussion

Mobile integrons are widespread genetic platforms involved in the interchange and expression of antibiotic resistance cassettes in bacteria. Antibiotic-induced cassette re-shuffling mediated by the SOS response (*Barraud and Ploy, 2015*; *Cambray et al., 2011*; *Guerin et al., 2009*) has been proposed to increase bacterial evolvability by providing 'adaptation on demand' to newly encountered antibiotics (*Engelstädter et al., 2016*; *Escudero et al., 2015*). We tested this hypothesis by

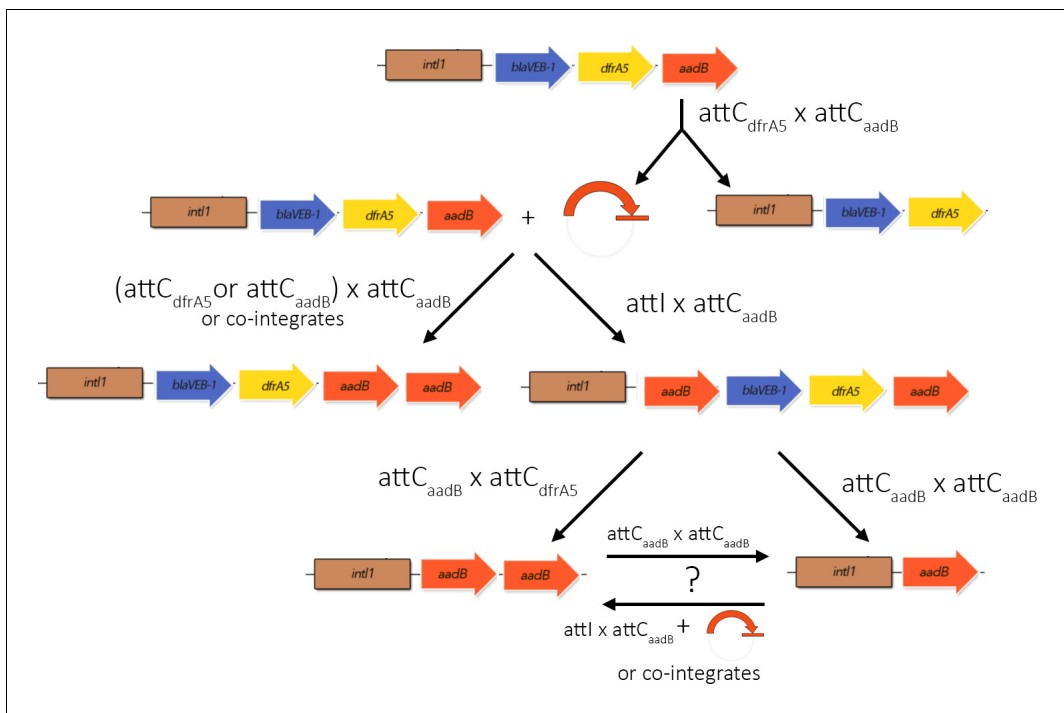

**Figure 6.** Representation of the proposed reactions leading to the various arrays observed in this study.

quantifying the impact of integrase activity on adaptation to increasing doses of gentamicin in populations of *P. aeruginosa* carrying a customized integron on a broad host range plasmid. Crucially, integrase activity accelerated the evolution of gentamicin resistance through rapid and repeated re-shuffling of the *aadB* resistance cassette, providing experimental support for the 'adaptation on demand' hypothesis.

We observed a diversity of cassette re-arrangements as a result of integrase activity, whose diverse prevalences can help us understand the evolutionary dynamics underlying cassette shuffling. Cassette shuffling and duplication were extremely frequent, both with the *aadB* and the *bla*$_{VIM-1}$ cassettes. The semi-conservative nature of cassette excision (*Escudero et al., 2015*) implies that cassette re-shuffling can lead to either 'cut and paste' or 'copy and paste' outcomes. Strikingly, all of the *aadB* re-arrangements that we observed were 'copy and paste' outcomes, resulting in the duplication of *aadB* (*Figure 6*). A bias towards evolution by 'copy and paste' is expected if increased copy number of the cassette under selection is beneficial. In this case, the benefit provided by *aadB* is strongly dependent on position, suggesting that 'copy and paste' insertions are unlikely to have provided stronger benefits than 'cut and paste' re-arrangements in first position. Furthermore, we only observe a slight advantage provided by *aadB-aadB* as compared to *aadB* arrays, suggesting that secondary copies of *aadB* were mostly redundant in arrays containing an *aadB* cassette in first position. The presence of multiple integrons within the same cell may also create an apparent bias towards 'copy and paste' re-shuffling: if re-insertion of an excised cassette is equally likely between all the integron copies present in a cell, we would expect the chance of an excised *aadB* cassette to re-insert into its original array to be 25–30% given the copy number of the R388 plasmid (2–3 per cell [*Fernández-López et al., 2006*]). However, the absence of any 'cut and paste' products in our experiments, instead of the expected 30%, suggests that the integrase enzyme may be inherently biased towards 'copy and paste' cassette re-arrangement, potentially by favoring the reinsertion of an excised cassette into the conserved array.

Although duplicated cassettes are relatively common in large chromosomal integrons, such as the *V. cholerae* superintegron (*Escudero et al., 2015*), they are rarely found in mobile integrons (*San Millan et al., 2015b*; *Stokes and Hall, 1992*). For example, duplicate cassettes are only found in 5% of the integrons that contain two or more cassettes in the INTEGRALL database (*Moura et al., 2009*; *Supplementary file 4*). Difficulties associated with resolving duplications from short-read

sequencing data probably contribute to this (*Alkan et al., 2011*), but it is clear that duplicate cassettes are rare. Interestingly, the duplicate copies of selected cassettes created by 'copy and paste' re-shuffling facilitate the loss of redundant cassettes. First, duplication of cassettes with highly recombinogenic *attC* sites, such as *aadB*, facilitates the integrase-mediated excision of redundant cassettes. In this case, the insertion of *aadB* in first position created the opportunity for the loss of the $bla_{VEB-1}$-*dfrA5* cassettes, through an $attC_{aadB} \times attC_{dfrA5}$ reaction, or the excision of $bla_{VEB-1}$-*dfrA5*-*aadB*, through an $attC_{aadB} \times attC_{aadB}$ recombination. Homologous recombination between duplicate cassettes or between copies of integrons on different plasmids provides a second mechanism for integron degeneration (*Andersson and Hughes, 2009*), in this case resulting in the formation of a single copy of the duplicate gene. This also highlights how extremely mobile cassettes, such as *aadB*, can compensate for the lack of mobility of other less recombinogenic cassettes, like $bla_{VEB-1}$ (*Aubert et al., 2012*). This bias toward integron degeneration may be detrimental over the long term by leading to the loss of potentially useful cassettes but may also promote the deletion of redundant cassettes with low excision rate, which could not otherwise be easily excised by the integrase. The constitutive expression of cassettes from the Pc promoter ensures that redundant cassettes impose a fitness cost (*Lacotte et al., 2017*), implying that selection for cassette loss is likely to be common. Given this, we argue that semi-conservative nature of cassette excision is key to the evolutionary benefits of integrase activity, as it allows integrons to rapidly gain additional copies of selected cassettes and facilitates the subsequent elimination of costly redundant cassettes.

In our experiments, mutations in the chromosome and in the integron made an important contribution to resistance evolution and are key to understanding the effective integrase evolutionary benefits. For example, integrase activity was associated with the loss of the redundant $bla_{VEB-1}$ cassette, which is in line with the integrase-mediated loss of redundant cassettes during selection for elevated chloramphenicol resistance observed by *Barraud and Ploy, 2015*. However, the $bla_{VEB-1}$ cassette was also rapidly inactivated by mutations in Δ*intI1* populations. While integrase activity offers additional evolutionary pathways to alleviate the cost of $bla_{VEB-1}$, the presence of numerous mutational targets achieving the same effect may offset the observable evolutionary benefit of the integrase. Similarly, mutations in the promoter of the *aadB* cassette provided an alternative mechanism to increase the expression of this cassette that did not depend on integrase activity. Moreover, *P. aeruginosa* has a strong potential to adapt to aminoglycosides via chromosomal mutations (*López-Causapé et al., 2018*; *Schurek et al., 2008*). Given our populations possess a sizable evolvability potential through mutations, even in the absence of integrase activity, we hypothesize an even stronger impact of the integrase should be observable in environments or species where evolution possibilities through mutations are more limited. Finally, we detected an interesting interplay between chromosomal evolution and integrase activity, with a divergence in the evolution of the two genotypes at the early $\times$4 MIC time point, which is potentially the most clinically relevant. Rapid chromosomal and plasmid evolution via mutations across a wide range of target genes allowed populations lacking a functional integrase to evolve resistance to increasing concentrations of gentamicin. The wild-type populations, on the other hand, showed a much-reduced mutation prevalence, as chromosomal mutations were potentially out-competed by the frequent and more efficient cassette re-arrangements. This accelerated chromosomal evolution in Δ*intI1A3* populations was short lived, confirming that rapid evolution was driven by more effective selection for mutations in populations lacking an integrase, rather than a difference in mutation rate per se. All populations that were able to survive very high levels of gentamicin exposure evolved by mutations of a common suite of target genes involved in antibiotic efflux and ribosomal modification, highlighting the fact that integrase activity did not ultimately alter the mutational routes to high-level resistance, but can impact chromosomal evolution during the early evolution of resistance.

Systems that upregulate the mutation rate under stress are widespread in bacteria (*Foster, 2011*; *MacLean et al., 2013*), and beneficial mutations generated by these systems can accelerate adaptation to stress (but see *Torres-Barceló et al., 2015*). However, most of the mutations generated by these systems will be deleterious, and stress-induced mutagenesis will therefore tend to reduce fitness, particularly if the deleterious effects of mutations are exacerbated by stress (*Kishony and Leibler, 2003*). The integron integrase is known to have off-target activity (*Recchia et al., 1994*), and it has been argued that the costs associated with off-target recombination contribute to the cost of integrase expression (*Harms et al., 2013*), limiting the evolutionary benefit of this system (*Engelstädter et al., 2016*). Although our sequencing strategy had limited ability to detect rare re-

arrangements (we imposed a cut-off of >30% prevalence) in any particular population, we compensated for this limitation by sequencing many replicate populations samples. We found no evidence of chromosomal re-arrangements or increased mutation rates that could be linked to integrase activity, with the exception of a single deletion in the R388 plasmid that was suggestive of integrase activity, suggesting a low rate of off-target recombination. It could be argued that the strong selection and regular population bottlenecking imposed by our experiment is likely to have been effective at purging populations from rare neutral and weakly deleterious variants produced by off-target integrase activity. However, a high rate of off-target integrase activity could have also resulted in an increased rate of fixation of mutations and/or re-arrangements, as a result of either positive selection (beneficial mutations produced by integrase activity) or hitch-hiking (neutral integrase-produced mutations linked to other beneficial mutations). In summary, our results support the idea that integrase activity accelerates evolution by creating high levels of variation exclusively in a region of the genome containing genes involved in response to stress, allowing bacteria to benefit from increased diversity without compromising genomic integrity. This contrasts with stress-induced mutagenesis, where the vast majority of mutations occur in genes that are not related to stress response, but further work is required to improve our understanding of the rate and spectrum of off-target recombination generated by integrase activity.

In conclusion, our study supports the view that integrons provide bacteria with an incredible opportunity to evolve in response to new antibiotic challenges by rapidly optimizing the expression of cassettes. Integrase activity allows bacteria to modulate cassette expression, rapidly gain additional copies of selected cassettes and eliminate redundant cassettes, while the high specificity of integrase-mediated recombination maintains genomic integrity, minimizing the costs of integrase activity. Given the importance of cassette re-shuffling, we argue that integrase activity will accelerate resistance evolution most for highly mobile cassettes that display strong positional effects, such as *aadB*. Given this, we argue that cassette re-shuffling will be most important in cases where bacteria have limited ability to adapt to antibiotics, for example when only a small number of mutations can increase resistance, or where resistance mutations carry large fitness costs. Integrase activity also provides bacteria with the opportunity to capture new resistance cassettes, an important challenge for future work will be to study the evolutionary processes driving cassette acquisition. Our work also supports the view that treatment strategies should seek to target integrons, for example by combining antibiotics with adjuvants that limit integrase activity by inhibiting the SOS response (*Hocquet et al., 2012*), or by using combinations of antibiotics that impose conflicting selective pressures on the integron.

# Materials and methods

**Key resources table**

| Reagent type (species) or resource | Designation | Source or reference | Identifiers | Additional information |
|---|---|---|---|---|
| Strain, strain background (*Pseudomonas aeruginosa*) | PA01 | Lab strain | NC_002516 | |
| Strain, strain background (*Escherichia coli*) | MG-1 | *Poirel et al., 1999* | AF205943 | *E. coli* clinical isolate containing a *qacI–aadB–aacA1/orfG–blaVEB1–aadB–arr2–cmlA5–blaOXA–10/aadA1* integron array |
| Strain, strain background (*Escherichia coli*) | EIEC-4 | *Gassama et al., 2004* | | *E. coli* clinical isolate containing a *dfrA5* integron cassette |

*Continued on next page*

*Continued*

| Reagent type (species) or resource | Designation | Source or reference | Identifiers | Additional information |
|---|---|---|---|---|
| Recombinant DNA reagent | R388 | *Avila and de la Cruz, 1988* | NC_028464.1 | |
| Recombinant DNA reagent | WTA1 | This study | | Custom integron array *drfA5– blaVEB1–aadB* on R388 plasmid backbone |
| Recombinant DNA reagent | WTA2 | This study | | Custom integron array *blaVEB1– aadB– dfrA5* on R388 plasmid backbone |
| Recombinant DNA reagent | WTA3 | This study | | Custom integron array *blaVEB1–dfrA5– aadB* on R388 plasmid backbone |
| Recombinant DNA reagent | WTA4 | This study | | Custom integron array *dfrA5– aadB–blaVEB1* on R388 plasmid backbone |
| Recombinant DNA reagent | WTA5 | This study | | Custom integron array *aadB–blaVEB1– dfrA5* on R388 plasmid backbone |
| Recombinant DNA reagent | WTA6 | This study | | Custom integron array *aadB–dfrA5– blaVEB1* on R388 plasmid backbone |
| Recombinant DNA reagent | Δint1A3 | This study | | Array WTA3 with a 948 bp deletion of the integrase *intI1* |
| Recombinant DNA reagent | pAMBL1 | *San Millan et al., 2015a* | KP873172.1 | Clinical plasmid containing a *blaVIM-1–aadB* integron array |
| Software, algorithm | breseq | *Barrick et al., 2014* | RRID:SCR_010810 | Version 0.33.2 |
| Software, algorithm | CNOGpro | *Brynildsrud, 2018* | | |

## Bacteria and growth conditions

A complete list of strains and plasmids can be found in *Supplementary file 1*. Unless stated, bacteria cultures were grown overnight at 37°C with shaking in Luria-Bertani (LB) Miller broth (Sigma Aldrich) and supplemented with 100 mg/L of ceftazidime when required to select for the integron-bearing plasmids.

## Strain construction

Six integron arrays covering all possible cassette orders were created using the plasmid R388 (*Avila and de la Cruz, 1988*) as backbone and three resistance cassettes: *aadB*, *bla*$_{VEB-1}$, and *dfrA5*.

The *bla*$_{VEB-1}$ and *aadB* cassettes were amplified from the integron of the *E. coli* MG-1 clinical isolate (***Poirel et al., 1999***), while the *dfrA5* cassette was obtained from an enteroinvasive *E. coli* strain isolated in Senegal (***Gassama et al., 2004***). These cassettes were then assembled into custom integron arrays using Gibson assembly and inserted into the plasmid R388 while replacing its original *dhfr–orf9–qacEΔ1–sul1* integron array (***Fernández-López et al., 2006***). The original R388 strong PcS promoter variant (high cassette expression but low integrase activity [***Jové et al., 2010***]) was replaced by the weaker PcW promoter from a clinical isolate to guarantee high integrase activity and represent the promoter most commonly found in class 1 integrons (***Jové et al., 2010***). Δ*intI1* mutants of these custom integrons were created by introducing a 948 bp deletion of the integrase *IntI1* gene during array construction, deleting most of the integrase open-reading frame but conserving both the Pint and Pc promoters. The final arrays were then first transformed into chemically competent *E. coli* MG1655.

These plasmids were then conjugated into *P. aeruginosa* through filter mating using the previous *E. coli* strains as donors and PA01 as recipient. Bacteria were incubated overnight in LB broth with 100 mg/L of carbenicillin at 37°C for the donors and in LB Miller broth without antibiotic at 42°C for the recipient bacteria. The next day cells were spun down, washed, and re-suspended in LB broth, before mixing in a 1:4 donor to acceptor ratio. The mix, as well as pure donor and acceptor controls, were put on filters placed on LB agar without antibiotics and incubated at 37°C overnight. Afterwards, filters were placed in tubes containing LB media and agitated. The resulting supernatants were plated on LB agar supplemented with 50 mg/L of kanamycin and 25 mg/L of ceftazidime and incubated for 48 hr. As *P. aeruginosa* PA01 has a higher innate resistance to kanamycin than *E. coli* MG1655 due to a chromosomally encoded phosphotransferase (***Okii et al., 1983***), kanamycin was used to select against the *E. coli* donors, while ceftazidime was used to select for the plasmid in the *P. aeruginosa* transconjugants. The final colonies were controlled by PCR for the presence of the plasmid and the absence of *E. coli* DNA.

## MIC determination

MIC were determined in cation-adjusted Mueller-Hinton Broth 2 (MH2), following the broth microdilution method from the Clinical and Laboratory Standards Institute guidelines (CLSI, 2017). Briefly $5\times10^5$ c.f.u bacteria inocula were prepared using individual colonies grown on selective agar in interlaced twofold-increasing concentrations of antibiotics and incubated for 20 hr in a shaking incubator at 37°C. The next day, plates' optical density ($OD_{595}$) was read using a Biotek Synergy two-plate reader. Wells were considered empty when the overall was under 0.1, and the MIC for each assay was defined as the minimal concentration in which growth was inhibited in all three technical replicates (separate wells, but grown on the same day from the same inoculum). The final MICs values are the average of two to four replicate assays (from separately prepared inocula, on different days).

## aadB cassette transcription levels
### RNA and DNA extractions

Each bacterial strain was inoculated in MH2 medium supplemented with antibiotics and grown overnight at 37°C with constant shaking (225 rpm). The overnight cultures were diluted 1:50 in fresh MH2 without antibiotics and incubated until they reached an $OD_{595}$ between 0.5 and 0.6. Both RNA and DNA were extracted for each sample. Half of each culture was mixed with RNAprotect Bacteria Reagent (Qiagen) according to the manufacturer instruction. Total RNA extraction was performed using the RNeasy Mini kit (Qiagen) on the QIAcube extraction machine (Qiagen). The other halves were treated with RNase and used to extract total gDNA using the DNeasy Blood and Tissue Kit on the QiaCube (Qiagen). Each strain was extracted three times from cultures started on different days.

### Plasmid copy number

Plasmid copy number was determined for each gDNA sample using the approach described in ***San Millan et al., 2015a***: all samples were first digested in order to linearize the plasmid using the restriction enzyme BamHI (BamHI FastDigest, ThermoFisher Scientific) according to the manufacturer instruction for gDNA digestion. Linearizing the plasmid increases DNA template accessibility and therefore prevents from underestimating the plasmid copy number (***Providenti et al., 2006***). The

amplified regions were controlled for the absence of BamHI restriction sites. The orf9 gene was used as the R388 plasmid target, and the mono-copy rpoD gene was used as chromosomal target for *P. aeruginosa* (primers given in *Supplementary file 1*). Amplifications were carried out using the Luna Universal Probe qPCR Master Mix (New England Biolabs). Thermal cycling protocol consisted of 1 s at 95°C (denaturation) and 20 s at 60°C (annealing/extension) for 40 cycles. Melting curve analysis was included for samples detected without probes. Fourfold dilution standard curves were included to control for differences in primer efficiencies. Plasmid copy number was calculated as the ratio between the plasmid and chromosomal target DNA quantities.

## Reverse transcription and qPCR

All RNA samples were treated with the TURBO DNA-free Kit (ThermoFisher) to eliminate genomic DNA. Concentration of the RNA samples was quantified using the Quantifluor RNA system (Promega). cDNA was synthesized from 100 ng of RNA templates using random primers from the GoScript Reverse Transcription Mix (Promega). qPCR was carried out on the StepOnePlus Real-time PCR platform (Applied Biosystems) using the iTaq Universal SYBR Green Supermix. The cassette, as well as two reference genes (actpA and acp), were amplified using the primers described in *Supplementary file 1* in two technical replicates for each extraction. Standard curves for the pair of cassette primers was included in each PCR using restriction enzyme-digested gDNA as template and used to quantify the amount of target cDNA in each sample to control for inter-run variations. Melting curve analysis was included after each run to test for non-specific amplification products. For each biological replicate, the cassette transcript levels were normalized based on the geometric means of the two internal reference genes, using the first array A1 as a reference, before division by its plasmid copy number.

## Experimental evolution with custom arrays

As antibiotics' MICs vary depending on the size of the starting inoculum (*Brook, 1989*), we first determined MIC in densities similar to the experimental evolution experiment (further called $MIC_{exp}$). Overnight cultures inoculated from two to three morphologically similar colonies grown on selective agar were incubated for 20 hr with shaking in MH2 media with antibiotics. These overnight cultures were then diluted 1/10,000 and supplemented with doubling concentrations of gentamicin in three replicates. $MIC_{exp}$ were determined the next day after 20 hr of incubation using $OD_{595}$ measurements. This process was repeated twice. In these conditions, the $MIC_{exp}$ for PA01:WTA3 and PAO1:$\Delta intI1$A3 were identical at 24 mg/L.

At the start of the experiment, 90 individual colonies grown on selective agar of each strain (PA01:WTA3 and PA01:$\Delta intI1$A3) were inoculated in 200 µL of MH2 media supplemented with gentamicin at a concentration of 1/8 $MIC_{exp}$. WT and $\Delta intI1$ strains were placed in a chequerboard pattern by interlacing the different genotypes to control easily for cross-contamination. Wells at the edge of every plate were kept bacteria free to avoid edge effects and identify contaminations. These 90 populations were passaged every day with a 1/10,000 dilution factor, and the antibiotic concentration was doubled, starting at 1/8 $MIC_{exp}$ until reaching a concentration of $\times$1024 $MIC_{exp}$. Alongside these 90 populations per strain which were transferred in increasing antibiotic concentrations, 30 populations per strain were transferred as controls in constant conditions: 15 without antibiotic and 15 at a constant concentration of 1/8 $MIC_{exp}$. Each population's $OD_{595}$ was measured each day and a population was considered extinct when its $OD_{595}$ fell below 0.1 after 20 hr of incubation. All populations were frozen in 15% glycerol every 2 days.

## DNA extraction

Liquid cultures were grown from the frozen stock of all surviving PA01:WTA3 and PA01:$\Delta intI1$ A3 populations at $\times$1024 $MIC_{exp}$ in LB Miller media supplemented with gentamicin at $\times$128 $MIC_{exp}$, and were incubated for 24 hr with shaking. Six populations were inoculated from each control treatment in either LB Miller supplemented with a concentration of $MIC_{exp}$ or with no antibiotic. For the $\times$4 $MIC_{exp}$ time point, 26 populations of each PA01:WTA3 and PA01:$\Delta$A3 genotype were regrown in $\times$2 $MIC_{exp}$ concentration of gentamicin. Ancestral PA01:WTA3 and PA01:$\Delta intI1$A3 populations were incubated in 100 mg/L of ceftazidime from the initial frozen stock. DNA extractions of the whole populations were carried out using the DNeasy Blood and Tissue Kit (Qiagen) on the QiaCube

extraction platform (Qiagen) combined with RNAse treatment. DNA concentrations were determined using the Quantifluor dsDNA system (Promega).

## PCR controls and analysis

At the ×1024 and ×4 MIC$_{exp}$ transfers all surviving populations were controlled for cross-contamination by verifying the size of the integrase by PCR (primers given in *Supplementary file 1*). Starting materials were either 2 µL of extracted DNA (×1024 MIC$_{exp}$ time point) or 2 µL of inoculate previously incubated for 24 hr and then boiled for 10 min at 105°C (×4 MIC$_{exp}$ time point). PCR were carried out using the GoTaq G2 DNA mastermix (Promega) and the following protocol: 30 s at 95°C, 30 s at 55°C, 3 min at 72°C for 30 cycles. Plate mishandling during the transfers resulted in the contamination of 40 and 34 wells of 90 ramping populations for each array. Areas of the plates where cross-contamination was detected in several wells in close proximity were excluded from the rest of the analysis, for a final population number of 65 per strain. The final populations at ×1024 MIC$_{exp}$ were analyzed by PCR to determine the position of the *aadB* cassette relative to the start and the end of the array as well as identify any *aadB* duplications or inversions and deletions of the plasmid backbone.

## Next-generation sequencing and bioinformatic pipeline

Library preparation and next-generation sequencing using the NovaSeq 6000 Sequencing System (Illumina) were carried out at the Oxford Genomics Centre at the Wellcome Centre for Human Genetics. Twenty-two PA01:WTA3 and 6 PA01:Δ*intl1*A3 populations were sequenced from the ×1024 MIC$_{exp}$ time point. For each control treatment, six populations were pooled together and sequenced as one. For the ×4 MIC$_{exp}$ time point, 26 PA01:WTA3 and 26 PA01:Δ*intl1*A3 were sequenced.

PCR duplications and optical artifacts were removed using MarkDuplicates (*Broad Institute, 2019*) and then low-quality bases and adaptors were trimmed from the sequencing reads using Trimmomatic v0.39 (*Bolger et al., 2014*). Finally, overall read quality control was performed using FastQC (*Simon Andrews, 2010*) and multiQC (*Ewels et al., 2016*). During this process, one PA01: WTA3 sample from the ×1024 MIC time point was removed due to the presence of non-Pseudomonas DNA. Average coverage depth was 760 (s.d. = 258) for the plasmid and 171 (s.d. = 24) for the chromosome.

Single-nucleotide polymorphism (SNP), point insertion, and deletion identification were performed using the breseq (*Barrick et al., 2014*) pipeline in polymorphism mode. For each population, reads were first mapped to the *P. aeruginosa* PA01 complete genome NC_ 002516.2 and the predicted sequence of WTA3. Non-mapped reads from the unevolved PA01:WTA3 population were then assembled de novo using SPAdes (*Bankevich et al., 2012*), and any open-reading frame identified using Prokka (*Seemann, 2014*) and further used as an additional reference to map unaligned reads from the other populations. The pipeline output was then further processed in MATLAB (MathWorks). Variants present in the unevolved ancestor populations at any frequency were filtered out. We also excluded variants reaching a frequency of less than 30% within a single population. A 5% threshold was applied to the pooled controls, which allows the detection of any variant present in more than 30% of a single population. Final results were exported in table format and processed for visualization using Circos (*Krzywinski et al., 2009*) and Geneious (Biomatters). Apart from the expected *intl1* deletion, the PA01:Δ*intl1*A3 genome was shown to differ from PA01:WTA3 and PA01 by two SNPs likely to have arisen during the conjugation process: one in PA3734 and one in the phzM/phzA1 intergenic region. PA3734 is predicted to be a lipase involved in cell-wall metabolism (*Dettman et al., 2015*) and may be involved in quorum-sensing (*Levesque, 2006*), while phzM and phzA1 are involved in the production of pyocyanins (*Higgins et al., 2018*). No literature linking those genes to aminoglycoside resistance was identified. Four PA01:Δ*intl1*A3 samples and one PA01: WTA3 sample from the ×4 MIC time point were removed from the analysis due to a wrong or mixed genotype from potential mislabeling or mishandling during DNA processing, leading to a final genomic dataset of 22 PA01:Δ*intl1*A3 and 24 PA01:WTA3 populations at ×4 MIC. All samples from the ×1024 MIC time point were of the correct genotype.

Potential new junctions between distant regions of the reference genome were identified through the *breseq* software on the plasmid and on the chromosome (*Barrick et al., 2014*). We detected

any re-arrangement present in more than 5% of a population, but kept only re-arrangements above 30% for further analysis, which excluded four re-arrangements, all deletions of less than 1 kb. Copy number variants were identified with CNOGpro (*Brynildsrud, 2018*) and used to confirm potential duplications or large-scale deletions. Finally, de novo assembly of the plasmids using plasmidSPAdes (*Antipov et al., 2016*) was carried out to provide additional evidence for the cassette rearrangements and visualized using Bandage (*Wick et al., 2015*). The robustness of the detection of cassette re-arrangement from the sequencing data was tested by cross-referencing the predicted recombinations with the results from the PCR screen at ×1024 MIC: all predicted recombinations matched the bands of the PCR screen, and only two extraneous bands could not be explained in three populations (*Figure 3—figure supplement 2*), confirming the robustness of the bioinformatic analysis.

## MIC determination of evolved populations

The MICs of 15 PA01:WTA3 and 5 PA01:Δ*intI1*A3 evolved populations from the final ×1024 MIC time point were determined as described above using individual colonies plated on in LB Miller media supplemented with gentamicin at ×128 MICexp from the frozen stocks.

To create populations containing the evolved plasmids in the ancestral chromosomal background, the plasmids of 15 PA01:WTA3 and 2 PA01:Δ*intI1*A3 ×1024 MIC populations containing a single type of array were first extracted using the QIAprep Miniprep (Qiagen) on the QiaCube extraction platform (Qiagen) from liquid culture in LB Miller supplemented with gentamicin at x64 MICexp. These plasmids were then electroporated back into the ancestral PA01 background using the protocol described in *Choi and Schweizer, 2006* and plated on agar supplemented with gentamicin at 24 mg/L. The presence of the plasmid was controlled by PCR targeting the integrase, and MICs were performed following the protocol described above.

## Experimental evolution with pAMBL1

Thirty colonies of PA01:pAMBL1 were inoculated in 100 µL of MH2 broth and transferred every day in doubling concentrations of meropenem with a 1/10,000 dilution and frozen in 15% glycerol every other day. Population survival was monitored each day after reaching a concentration of ×1 MIC$_{exp}$ by plating every well on a MH2 agar plate without antibiotic using a pin replicator. Extinction of a population was defined as the absence of a visible colony after 24 hr incubation at 37°C.

Surviving populations at ×2 MIC$_{exp}$ were grown on a MH2 agar plate without antibiotic and used as substrate for PCR. Primers were used to identify potential duplications of the bla$_{VIM}$ cassette by PCR. PCR were carried out using the GoTaq G2 DNA mastermix (Promega) and the following protocol: 30 s at 95°C, 30 s at 55°C, 3 min at 72°C for 30 cycles. Single clones from two different populations were sequenced through whole genome sequencing and analyzed using the same protocol as described previously.

## Statistical analysis

Statistical analysis was carried out using R (version 3.6.1) and RStudio (Version 1.2.5). Survival analysis using the log-rank test was performed using the survival (*Therneau, 2020*) package to compare survival rates between populations with and without a functional integrase.

## Acknowledgements

We thank Laurent Poirel and Marie-Cecile Ploy for the gift of the strains containing the cassettes used in the construction of the integrons arrays. We are grateful to Natalia Kapel and Gerda Kildisiute for experimental support and Julio Diaz Caballero and Jessica Hedge for bioinformatic support. This project was funded by Wellcome Trust Grant 106918/Z/15/Z held by RCM. We thank the Oxford Genomics Centre at the Wellcome Centre for Human Genetics (funded by Wellcome Trust grant reference 203141/Z/16/Z) for the generation and initial processing of the sequencing data. CS was supported by funding from the Biotechnology and Biological Sciences Research Council (BBSRC) [grant number BB/M011224/1]. JAE was supported by the European Research Council (ERC) through a Starting Grant (803375), the *Atracción de Talento* Program of the *Comunidad de Madrid* (2016-T1/BIO-1105) and the *Ministerio de Ciencia, Innovación y Universidades* (BIO2017-85056-P)

## Additional information

### Funding

| Funder | Grant reference number | Author |
|---|---|---|
| Wellcome Trust | 106918/Z/15/Z | R Craig MacLean |
| Biotechnology and Biological Sciences Research Council | BB/M011224/1 | Célia Souque |
| H2020 European Research Council | 803375 | José Antonio Escudero |
| Comunidad de Madrid | 2016-T1/BIO-1105 | José Antonio Escudero |
| Ministerio de Ciencia, Innovación y Universidades | BIO2017-85056-P | José Antonio Escudero |

The funders had no role in study design, data collection and interpretation, or the decision to submit the work for publication.

### Author contributions

Célia Souque, Conceptualization, Formal analysis, Validation, Investigation, Visualization, Methodology, Writing - original draft, Writing - review and editing; José Antonio Escudero, Conceptualization, Resources, Supervision, Investigation, Methodology, Writing - review and editing; R Craig MacLean, Conceptualization, Resources, Supervision, Funding acquisition, Writing - original draft, Project administration, Writing - review and editing

### Author ORCIDs

Célia Souque (ID) https://orcid.org/0000-0001-7194-4322
José Antonio Escudero (ID) https://orcid.org/0000-0001-8552-2956

### Decision letter and Author response

Decision letter https://doi.org/10.7554/eLife.62474.sa1
Author response https://doi.org/10.7554/eLife.62474.sa2

## Additional files

### Supplementary files

- Supplementary file 1. Primers used in this study.
- Supplementary file 2. List of mutations / recombinations at x1024 MIC.
- Supplementary file 3. List of mutations / recombinations at x4 MIC.
- Supplementary file 4. List of duplicated cassettes from the INTEGRALL database.
- Transparent reporting form

### Data availability

Sequencing data have been deposited on ENA under the accession code PRJEB40301 Source data files have been deposited on Dryad for Figures 1,2,3 and Figure 1—figure supplement 1. All other data generated or analysed during this study are included in the manuscript and supporting files.

The following datasets were generated:

| Author(s) | Year | Dataset title | Dataset URL | Database and Identifier |
|---|---|---|---|---|
| Souque Cl, Escudero JA, MacLean RC | 2020 | Adaptive benefits of integron shuffling | https://www.ebi.ac.uk/ena/browser/view/PRJEB40301 | European Nucleotide Archive, PRJEB40301 |
| Souque Cl, Escudero JA, | 2020 | Integron array MICs and cassettes transcription data | https://doi.org/10.5061/dryad.rv15dv469 | Dryad Digital Repository, 10.5061/ |

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
