## [Decision Letter]

**Acceptance summary:**

In this study, the authors test the long-standing hypothesis that integrons promote "evolution-on-demand". Using a combination of custom integrons and experimental evolution examining resistance to gentamycin, the authors present a convincing body of work supporting this hypothesis.

**Decision letter after peer review:**

Thank you for submitting your work entitled "Integron activity accelerates the evolution of antibiotic resistance" for consideration by *eLife*. Your article has been reviewed by three peer reviewers, and the evaluation has been overseen by a Reviewing Editor and Gisela Storz as the Senior Editor.

In this manuscript, the authors test the long-standing "evolution-on-demand" hypothesis of integrons. Using a combination of genetic construction work, experimental evolution, and WGS the authors present a convincing body of work favoring the presented hypothesis. They introduce three antibiotic gene cassettes into an integron and use an "evolutionary ramp" approach with gentamicin and demonstrate that the gentamicin resistance cassette shuffles towards the start of the integron. This provides compelling evidence favoring the evolutionary effects of an active class 1 integrase.

The paper is clear, well written and demonstrates neatly the benefits of integron shuffling. The authors should also be given credit for including experimental data from an integron containing a clinical plasmid including resistance cassettes to the last resort antibiotics carbapenems. This is largely missing in the field.

Our overall assessment of the manuscript is positive. However, a number of questions have been raised regarding the mechanistic aspects and conclusions of the paper. We are therefore suggesting additional assays to measure phenotypic effects of evolved integrons, and data analyses on the negative controls. If these are not possible to perform, the main conclusions could be slightly altered instead to focus more on the genomics. Finally, we provide some suggestions on making the Discussion more balanced and in clarifying the role of chromosomal mutations in the integron-facilitated evolution.

For the manuscript to be accepted, the following points should be addressed:

1) The paper is framed around the evolution of increased antibiotic resistance, which is not directly quantified. Assays to measure resistance of evolved clones (from the populations in which integrons are not polymorphic (Figure 3B)) would demonstrate and quantify the evolution of increased resistance associated with the different integron arrays (aadB-bla_VEB-1_-dfrA5-aadB, aadB-aadB, aadB and the ancestral array), allowing to link genotype and phenotype. By transferring evolved plasmids to an unevolved host, effects of plasmid evolution could also relatively easily be separated from chromosomal mutations that may contribute to gentamicin resistance. If these additional experiments are not possible, the text should be reformulated to focus on the genetics and not on phenotypic resistance.

2) One question that came up was whether there were any re-arrangements in the negative control that was evolved without antibiotics. After discussion, we concluded that the absence of this analysis was "because the entire R388 plasmid was lost in every control population that was passaged in antibiotic-free medium". It may be worth highlighting this plasmid loss and more generally clarifying the results of the different controls (1/8 MIC and no antibiotic).

3) What sequence coverage was used? Could it be that random re-arrangements appear at a very low frequency that are only fixed under changing conditions (similar to mutations)?

4) Explain better Figure 3—figure supplement 2, showing the crucial control where PCRs confirm data from Illumina short read sequencing on whole populations. A schematic figure of each combination of cassettes, primer positions, and expected band length combined with proper lane descriptions should be prepared in a revised version of the manuscript.

5) The authors argue that intI1 has a bias towards "copy and paste" as opposed to "cut and paste" cassette rearrangements. Why use the term "copy" and how is this novel? In the two outcomes in integron structure, the cassette is excised (cut) from the ancestral integron before it is inserted (paste) into either arrays. Moreover, is there sufficient evidence to support the proposed "copy and paste" bias of IntI1? As the authors discuss thoroughly, the presence of multiple copies of the ancestral structure provides more "ancestral" integration targets for the excised cassettes. The authors exclude the alternative hypothesis that a second copy of aadB increased fitness as compared to a single copy (as expected from copy and paste). Measuring the fitness effects of different arrays (point 1) would address this point.

6) The authors highlight that they found no evidence of deleterious off-target integrase effects. They suggest that integrase activity may purge deleterious chromosomal mutations and enable more targeted beneficial adaptive responses. The authors present cases where likely beneficial off target recombination events occurred. To what extent do the authors think the absence of deleterious off target effects is due to the experimental conditions (continuous increments in gentamicin concentrations combined with strong bottlenecks)?

---

## [Author Response]

[…] Our overall assessment of the manuscript is positive. However, a number of questions have been raised regarding the mechanistic aspects and conclusions of the paper. We are therefore suggesting additional assays to measure phenotypic effects of evolved integrons, and data analyses on the negative controls. If these are not possible to perform, the main conclusions could be slightly altered instead to focus more on the genomics. Finally, we provide some suggestions on making the Discussion more balanced and in clarifying the role of chromosomal mutations in the integron-facilitated evolution.For the manuscript to be accepted, the following points should be addressed:1) The paper is framed around the evolution of increased antibiotic resistance, which is not directly quantified. Assays to measure resistance of evolved clones (from the populations in which integrons are not polymorphic (Figure 3B)) would demonstrate and quantify the evolution of increased resistance associated with the different integron arrays (aadB-bla_VEB-1_-dfrA5-aadB, aadB-aadB, aadB and the ancestral array), allowing to link genotype and phenotype. By transferring evolved plasmids to an unevolved host, effects of plasmid evolution could also relatively easily be separated from chromosomal mutations that may contribute to gentamicin resistance. If these additional experiments are not possible, the text should be reformulated to focus on the genetics and not on phenotypic resistance.

In response to this helpful suggestion, we have carried out two further experiments.

a) First, we measured the resistance of evolved populations, allowing us to quantify the extent of resistance evolution during our selections. Importantly, we show that all WT and ∆intI1 populations evolved to similar levels of gentamicin resistance (MIC= 25g/L). The methods for this experiment are described in the text and the results are represented in Figure 2C and reported in the text.

b) Second, we measured the resistance level provided by evolved integrons by transforming evolved plasmids into a PA01 reference strain. This experiment allowed us to confirm that the integron evolved to provide large increases in resistance in all populations. The methods for this experiment are described in the text and the results are represented in Figure 3E and reported in the text.

2) One question that came up was whether there were any re-arrangements in the negative control that was evolved without antibiotics. After discussion, we concluded that the absence of this analysis was "because the entire R388 plasmid was lost in every control population that was passaged in antibiotic-free medium". It may be worth highlighting this plasmid loss and more generally clarifying the results of the different controls (1/8 MIC and no antibiotic).

In response to this helpful point, we have highlighted the loss of the plasmid in the antibiotic-free control populations and added sentences describing the chromosomal mutations in the controls. We also highlight the plasmid mutations and the rearrangement of the 1/8 MIC controls. (Results).

3) What sequence coverage was used? Could it be that random re-arrangements appear at a very low frequency that are only fixed under changing conditions (similar to mutations)?

We agree with the reviewer that our sequencing has limited ability to detect very rare integron re-arrangements. Our depth of sequencing coverage was high (760 +/- 258 for the plasmid, 171 +/- 24 for the chromosome) and while we restricted our analysis to variants present in a least 30% of a single population, our bioinformatic pipeline should be able to detect any re-arrangements with a frequency greater than about 5%. We only detected 4 rearrangements that did not reach the 30% threshold, equally divided between both genotypes, and all were deletions under 1kb. Given the uncertainties regarding these rare rearrangements, we did not include them in our manuscript. One strength of our experimental design is that we sequenced a large number of populations with a functional integrase, and this high level of replication is key to our conclusion that off-target effects of the integrase were rare (Results).

We have added information on coverage for each population in Supplementary files 2 and 3, and indicated average sequence coverage in the Materials and methods. We also added a sentence about our lower bound for rearrangement detection.

4) Explain better Figure 3—figure supplement 2, showing the crucial control where PCRs confirm data from Illumina short read sequencing on whole populations. A schematic figure of each combination of cassettes, primer positions, and expected band length combined with proper lane descriptions should be prepared in a revised version of the manuscript.

We have revised this important supplementary figure showing confirmatory data along the lines suggested by the reviewer.

5) The authors argue that intI1 has a bias towards "copy and paste" as opposed to "cut and paste" cassette rearrangements. Why use the term "copy" and how is this novel? In the two outcomes in integron structure, the cassette is excised (cut) from the ancestral integron before it is inserted (paste) into either arrays. Moreover, is there sufficient evidence to support the proposed "copy and paste" bias of IntI1? As the authors discuss thoroughly, the presence of multiple copies of the ancestral structure provides more "ancestral" integration targets for the excised cassettes.

We agree with the reviewer that “copy and paste” and “cut and paste” reflect the fact that the same process (integron cassette excision and integration) can have two different outcomes, and we have clarified this point in the revised manuscript (Introduction and Discussion). We also agree with the reviewer that, all else being equal, the multi-copy nature of the integron should create a statistical bias towards observing “copy and paste”. Given the copy number of the integron in our system, we argue that our default expectation should be a 75% of “copy and paste” (Discussion). What is remarkable is that we observe “copy and paste” in 100% cases (n=14 of populations with a simple aadB insertion at x1024 MIC), suggesting an inherent bias towards “copy and paste”, as the chance of never seeing the insertion of the excised copy in the ancestral array in 14 populations is around (3/4)^14^~1.9%.

The authors exclude the alternative hypothesis that a second copy of aadB increased fitness as compared to a single copy (as expected from copy and paste). Measuring the fitness effects of different arrays (point 1) would address this point.

We thank the reviewer for this helpful point, as a similar fitness between the two arrays is necessary in our reasoning. In the revised manuscript, we show that evolved integrons carrying duplicate copies of aadB from a “copy and paste” outcome have very similar MICs to the ancestral WTA5 array, which would have come from “cut and paste” outcome. This is not a perfect comparison, because the evolved “copy and paste” arrays also carry mutations in bla_VEB-1_, but is clear that any differences in resistance associated with “copy and paste” and “cut and paste” outcomes must be small (Figure 3E).

6) The authors highlight that they found no evidence of deleterious off-target integrase effects. They suggest that integrase activity may purge deleterious chromosomal mutations and enable more targeted beneficial adaptive responses.

The reviewers’ point here is somewhat unclear to us. Deleterious mutations are purged from populations by selection, not integrase activity. Rather, we have tried to contrast stress-induced integrase activity, which generates large variation in a small region of the genome, with stress-induced mutagenesis, which creates variation across the entire genome (Discussion).

The authors present cases where likely beneficial off target recombination events occurred. To what extent do the authors think the absence of deleterious off target effects is due to the experimental conditions (continuous increments in gentamicin concentrations combined with strong bottlenecks)?

In the revised manuscript we highlight that we only uncovered a single instance of potential off target recombination. Although theoretical considerations suggest that our experimental design may have limited our ability to detect rare neutral or mildly deleterious variants produced by off-target recombination, it also opened up the possibility for beneficial off-target variants to spread. Thus, it is unclear to what extent our experimental design biased our detection of off-target recombination, and we have emphasized the need to further understand the off-target effects of the integron (Discussion).